∂ | **Open Peer Review** | Antimicrobial Chemotherapy | Observation

# Guanabenz acetate, an antihypertensive drug repurposed as an inhibitor of *Escherichia coli* biofilm

Arakkaveettil Kabeer Farha,[1] Olivier Habimana,[1,2] Harold Corke[1,2]

**ABSTRACT** Biofilms formed by *Escherichia coli* are composed of amyloid curli and cellulose and have been shown to be linked to pathogenicity, antibiotic resistance, and chronic infections. Guanabenz acetate (GABE), an antihypertensive drug, was identified as a potential strategic repurposing drug due to its biofilm inhibitory properties following an extensive antimicrobial screening assay of 2,202 Food and Drug Administration-approved non-antibiotic agents. The results of this study provide insights into the effectiveness of GABE as a therapeutic alternative against *E. coli* biofilm-associated infectious diseases.

**IMPORTANCE** Biofilm-associated bacterial infections are one of the major problems in medical settings. There are currently limited biofilm inhibitors available for clinical use. Guanabenz acetate, a drug used to treat high blood pressure, was found to be an effective anti-biofilm agent against *Escherichia coli*. Our results show that this drug can inhibit the production of cellulose and curli amyloid protein, which are the two main components of *E. coli* biofilms. Our findings highlight the possibility of repurposing a drug to prevent *E. coli* biofilm formation.

**KEYWORDS** guanabenz acetate, curli, biofilm inhibitor, *Escherichia coli*, drug repurposing

B acteria are known for forming a well-organized structure of a self-produced matrix of extracellular polymeric substances (EPS) composed of proteins, exopolysaccharides, and extracellular DNA. The biofilm's EPS provide its embedded cells with a first line of defense against environmental stresses through a cascade of mechanisms involving the prompting of antibiotic resistance, nutrient dispersal, and gradual release of signaling molecules within the community (1). However, due to the difficulty of biofilm removal and the resilience exhibited by the bacteria within them, biofilms produced by pathogenic bacteria pose severe threats in a clinical context through their adaptation and survival within the clinical environments. The acquisition of antibiotic and antimicrobial resistance by pathogen-associated biofilms can lead to severe consequences in patients suffering from nosocomial and chronic wound infections (2). New alternatives to antibiofilm strategies have recently been a key focus of research to curb the spread of antibiotic-resistant organisms within clinical settings; one strategy is the search for efficient and potent biofilm inhibitor drugs.

The biofilms of enteric bacteria, such as *Escherichia coli* and *Salmonella* spp., are mainly composed of two components: curli amyloid fibers, which are proteinaceous, and cellulose (3). Curli are responsible for the first attachment step in the early stages of biofilm development. Additionally, curli can interact with different host proteins by triggering an inflammatory response in the host and are implicated in pathogenesis (4). A drug-repurposing approach was employed to screen 2,202 Food and Drug Administration-approved non-antibiotics using *E. coli* AR3110, a non-pathogenic *E. coli* K-12 W3110

Address correspondence to Harold Corke, harold.corke@gtiit.edu.cn, or Arakkaveettil Kabeer Farha, farha.kabeer@gtiit.edu.cn.

The authors declare no conflict of interest.

See the funding table on p. 5.

strain derivative, as a biofilm model to identify enteric biofilm inhibitors. *E. coli* AR3110 consists of curli fibers and pEtN cellulose in the biofilm matrix with long radial ridges and small wrinkles that stain dark red with Congo red when grown on substrates and show pellicle phenotype when grown in liquid culture (5, 6). We took advantage of the ability of *E. coli* AR3110 to form different biofilm phenotypes to design the screening assay for biofilm studies. Guanabenz acetate (GABE) (Fig. 1A) was identified as a potent biofilm inhibitor in the screen. GABE is an antihypertensive drug that can act as an alpha-2 adrenergic receptor agonist (7). Recently, GABE has been proven to display chemopreventive properties (8), obesity-associated fatty liver and hyperglycemia (9), and Alzheimer's disease-associated neuropathological alteration inhibitory activities (10). GABE altered the phenotype of *E. coli* AR3110 in Congo red agar to a smooth and white color colony showing a lower uptake of Congo red (Fig. 1B; Fig. S1) and also reduced the thickness of pellicle in liquid culture (Fig. 1C). Additionally, the curli production of the *E. coli* K12 W3110 strain, which decreases in curli production due to GABE treatment, was also observed (Fig. S1).

Using the microbroth dilution method, the minimal inhibitory concentration and the minimum bactericidal concentration of GABE against three tested *E. coli* strains, namely, *E. coli* MG1655, *E. coli* AR3110, and *E. coli* W3110, were found to be 1,000 and 2,000 µM, respectively. Despite a slight suppression of bacterial growth, growth curve and spot assay analysis showed that at 200 µM, GABE did not impair the bacterial viability (Fig. 1E; Fig. S2). *E. coli* AR3110 grown on a calcofluor-supplemented LB salt-free agar with GABE showed very weak fluorescence, confirming the inhibition of cellulose production. In contrast, the control showed strong fluorescence (Fig. 1D). Additionally, the submerged biofilm formation by *E. coli* AR3110 was highly inhibited by GABE as identified by crystal violet staining (Fig. 1F), slightly affecting bacterial viability (Fig. 1G; Fig. S2).

Confocal laser scanning microscopy analysis showed the presence of abundant biofilms in control *E. coli* AR3110 attached on the glass slide, as indicated by high green fluorescence. The thickness of the biofilms was significantly reduced with GABE treatment (Fig. 1H). The scanning electron microscopy image of the control revealed a biofilm enriched with EPS matrix and densely packed bacteria. After treatment with GABE, very few cells remained on the glass slide. A lack of EPS was observed, indicating the effectiveness of GABE as a biofilm inhibitor (Fig. 1I). Thioflavin T fluorescence intensity was not observed in GABE-treated wells, while increased fluorescence was observed in the control, indicating that GABE inhibited the polymerization of CsgA *in vitro* (Fig. S3). As shown in Fig. 1J, GABE suppressed curli production, as visualized by transmission electron microscopy (TEM).

An RNA transcriptomic study was conducted to investigate the possibility that GABE can decrease curli biogenesis at the transcription level. According to transcriptome profiling, 763 significantly differentially expressed genes (DEG) were found (Fig. 1K). The suppression of genes in *csg* operons (*csgA*, *csgB*, *csgC*, *csgD*, *csgE*, *csgF*, and *csgG*) was found to be consistent with the reported smooth and white biofilm phenotype of *E. coli* AR3110 (11). Many of the genes with decreases in expression in response to the GABE challenge were related to acid resistance (AR) systems, including AR1 (*hdeAB* operon) and AR2 (gad system) (12) (Table S1). Similar effects of downregulation of *hdeA*–*hdeB* have been shown when the *E. coli* biofilms were exposed to thymol, carvacrol, ursolic acid, and chlorhexidine–digluconate biocide (13–15). This might suggest a lower acid resistance to bacteria at acidic pH. Though the acetate form of GABE is effective in clinical settings, GABE elicits a reaction opposite to the acetate-induced stress response, where *E. coli* acid stress response enhances bacterial survival (16). The reversal of the acid stress response could be attributed to aminoguanidine's fundamental nature rather than the anionic counter ion (17). The propanoate metabolism-related *prpCDE*, which promotes *E. coli* biofilm formation, was significantly downregulated in GABE-treated cells compared to the control (18). Following exposure to GABE, *torCAD* operon, which is linked to anaerobic respiration, and genes associated with YbhFSR putative drug resistance exporter were substantially elevated. The current work confirms prior reports of the

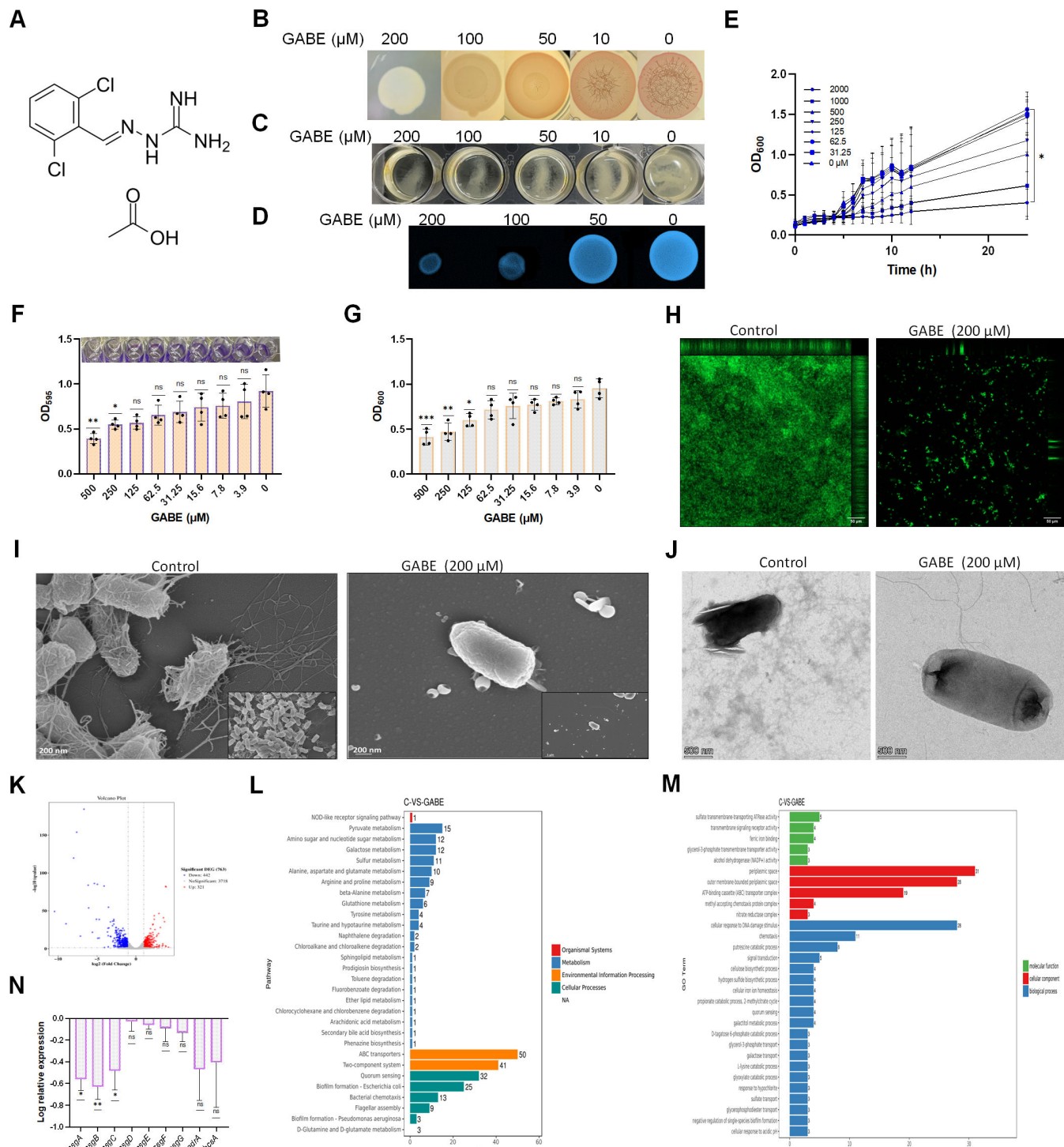

**FIG 1** *Escherichia coli* AR3110 biofilm inhibition effect of GABE. (A) Structure of guanabenz acetate (GABE). (B) Colony morphology of *E. coli* AR3110 on Congo red agar. *E. coli* AR3110 (2 µl) was spotted on lysogeny broth (LB) salt-free agar supplemented with Congo red and Coomassie brilliant blue, achieving final concentrations of 40 and 20 µg/ml, respectively. The stock solution was prepared by dissolving 50 mg GABE in 1 mL 100% dimethyl sulfoxide (DMSO) (171.74 mM). In our experiments, the final concentration of DMSO was only 0.1%. The plates were incubated with or without GABE at 28°C for 96 h. The colony morphology was visualized by stereomicroscope (10× magnification). (C) Pellicle mode of the *E. coli* AR3110 biofilm inhibition by GABE. *E. coli* AR3110 was grown in LB salt-free broth in 24-well plates with or without GABE for 96 h at 28°C under static conditions. At the end of the incubation period, pellicle was visually identified. The image was taken by a phone camera (iPhone 14 Promax). (D) Calcofluor staining of the *E. coli* AR3110 biofilm. *E. coli* AR3110 was grown on LB salt-free agar supplemented with calcofluor stain (200 µg/ml) with or without GABE for 48 h at 28°C under static conditions. The fluorescence was observed

**Fig 1 (Continued)**

under UV light (365–395 nm wavelength). The image was captured by a gel-imaging instrument (Fusion FX7 EDGE, Vilber). (E) Growth curve of *E. coli* AR3110. *E. coli* AR3110 was grown in LB media with various concentrations of GABE without shaking at 37°C for 24 h, and OD at 600 nm was measured every hour interval until 12 h and 24 h. Statistical significance was determined with Kruskal–Wallis test, followed by Dunn's multiple comparison test. Data are expressed as mean ± standard deviation ($n = 3$), with asterisks indicating the level of statistical significance: *$P < 0.05$, **$P < 0.01$, and ns = non-significant. (F) Biofilm quantification by crystal violet staining. The inset shows an image of a 96-well biofilm assay plate. (G) Biomass analysis. *E. coli* AR3110 was grown in LB salt-free broth with or without GABE for 48 h at 28°C under static conditions. The OD value was measured at 600 nm. Then, the planktonic cells were removed, and the biofilms were stained with crystal violet. OD at 595 was measured after dissolving crystal violet in 95% ethanol. The assay was performed in four biological replicates and technical duplicates in each experiment. Statistical significance was determined with Kruskal–Wallis test, followed by Dunn's multiple comparison test. Data are expressed as mean ± standard deviation, with asterisks indicating the level of statistical significance: *$P < 0.05$, **$P < 0.01$, and ns = non-significant. (H) Confocal laser scanning microscopy images of the biofilm. *E. coli* AR3110 was grown on a coverslip immersed in LB salt-free broth with or without GABE for 48 h at 28°C under static conditions. The biofilms were stained with SYTO-9 (485/498 nm), and images were taken. (I) Scanning electron microscopy images of the *E. coli* AR3110 biofilm grown on a coverslip in LB salt-free broth with or without GABE for 48 h at 28°C under static conditions. (J) Transmission electron microscopy images showing the inhibition of curli in GABE-treated *E. coli* AR3110 biofilm. *E. coli* AR3110 was grown on LB salt-free agar with or without GABE for 48 h at 28°C under static conditions, negatively stained with uranyl formate. (K) Volcano plot for differentially expressed genes obtained by RNA transcriptomic data analysis. (L) Gene Ontology enrichment analysis. (M) Kyoto Encyclopedia of Genes and Genomes pathway enrichment analysis. (N) Quantitative real-time polymerase chain reaction (PCR) data of *csg* operon genes and genes associated with cellulose synthesis. For transcriptomic, as well as real-time PCR analysis, *E. coli* AR3110 was grown in LB salt-free broth with or without GABE for 48 h at 28°C under static conditions. Student's *t*-test was used to analyze the significant difference between control and treated samples. All experiments were performed independently three times. Data are expressed as mean ± standard deviation, with asterisks indicating the level of statistical significance: *$P < 0.05$, **$P < 0.01$, and ns = non-significant.

overexpression of *torCAD* operon and ATP-binding cassette (ABC) transporters in biofilms following drug treatment (19, 20). Pathway enrichment analysis revealed that GABE had an impact on several pathways related to the regulation of biofilm development, such as ABC transporters, quorum sensing, two-component systems, and biofilm formation in *E. coli* (21) (Fig. 1L and M).

Cellulose synthesis occurs only when diguanylate cyclase C, a putative transmembrane protein regulated by CsgD, is expressed. Consistent with the calcofluor staining result, GABE altered the cellulose biosynthesis genes by downregulating many genes involved in cellulose synthesis (Table S1). Motility experiments revealed that GABE slightly affected *E. coli* AR3110 swarming motility but had no effect on swimming motility (Fig. S4). GABE treatment decreased the expression of multiple genes linked to *E. coli* motility (Supplementary File 2). The transcriptional changes associated with the operons *csgBAC*, *csgDEFG*, *adrA*, and *bcsA* were confirmed by quantitative real-time polymerase chain reaction (Fig. 1N). The CsgA protein was dramatically decreased with GABE, which is fully consistent with the TEM observation. Moreover, the expression of the CsgD protein, the master regulator of the curli operon, was also decreased (Fig. S5).

In summary, our study demonstrates the antibiofilm effectiveness of GABE, which will be further investigated in clinically important strains and preclinical models to confirm the compound's potential for clinical application. Drugs with antimicrobial activity, but not intended for use as antibiotics, are classified as non-antibiotics (22). The absence of the bactericidal effect of an antibiofilm agent reduces the selective pressure against biofilm development, hence reducing the likelihood of bacteria developing resistance to it (23). At lower doses, GABE can demonstrate an antibiofilm effect without impeding bacterial growth. In this context, it is anticipated that bacteria will be less likely to become resistant to GABE, a non-antibiotic drug that inhibits biofilms. It has been proposed that certain hypertension drugs, including propranolol, doxazoin, and methyl-L-DOPA, can be effectively used as antibacterial drugs in mice (24–26). Similarly, GABE has the potential to be a practical therapeutic alternative to lessen chronic illnesses in humans linked to curli biofilms. Studies on the adverse effects of antihypertensive drugs used to treat bacterial infections are limited. Even if GABE is already safe for human use, it should be necessary to reevaluate its pharmacological properties to develop new therapeutic applications.

## ACKNOWLEDGMENTS

We are grateful to Prof. Regine Hengge (Institut für Biologie/Mikrobiologie, Humboldt-Universität zu Berlin, Germany) for providing the *Escherichia coli* AR3110 strain. We thank AZENTA Life Sciences (Suzhou, China) for the next-generation sequencing services.

This research was supported by the Research Fund for International Young Scientists (RFIS-I), the National Natural Science Foundation of China (82150410458) (to A.K.F.), and Foreign Expert Project 2022 (to A.K.F.).

## AUTHOR AFFILIATIONS

[1]Biotechnology and Food Engineering Program; and Key Laboratory of Science and Engineering for Health and Medicine of Guangdong Higher Education Institutes, Guangdong Technion-Israel Institute of Technology, Shantou, China

[2]Faculty of Biotechnology and Food Engineering, Technion-Israel Institute of Technology, Haifa, Israel

## AUTHOR ORCIDs

Arakkaveettil Kabeer Farha http://orcid.org/0000-0001-7270-1461

## FUNDING

| Funder | Grant(s) | Author(s) |
| --- | --- | --- |
| MOST \| National Natural Science Foundation of China (NSFC) | 82150410458 | Arakkaveettil Kabeer Farha |

## AUTHOR CONTRIBUTIONS

Arakkaveettil Kabeer Farha, Conceptualization, Formal analysis, Funding acquisition, Investigation, Methodology, Project administration, Validation, Writing – original draft, Writing – review and editing | Olivier Habimana, Formal analysis, Validation, Writing – review and editing | Harold Corke, Formal analysis, Project administration, Supervision, Validation, Writing – review and editing

## DATA AVAILABILITY

The raw RNA sequencing data are available under BioProject ID PRJNA1073484.

## ADDITIONAL FILES

The following material is available online.

### Supplemental Material

**Supplemental material (Spectrum00738-24-s0001.doc).** Table S1; Fig. S1 to S5.
**Supplemental data (Spectrum00738-24-s0002.xlsx).** Significant DEGs identified in RNA transcriptomic analysis.

### Open Peer Review

**PEER REVIEW HISTORY (review-history.pdf).** An accounting of the reviewer comments and feedback.

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
