## [Reviewer comments · Microbiology Spectrum]

Microbiology Spectrum

Guanabenz acetate, an antihypertensive drug repurposed as an inhibitor of *Escherichia coli* biofilm

Arakkaveetil Farha, Olivier Habimana, and Harold Corke

Corresponding Author(s): Arakkaveetil Farha, Guangdong Technion-Israel Institute of Technology Department of Biotechnology and Food Engineering

Review Timeline:

Submission Date:	March 21, 2024
Editorial Decision:	June 11, 2024
Revision Received:	July 31, 2024
Editorial Decision:	August 2, 2024
Revision Received:	August 6, 2024
Accepted:	August 9, 2024

Editor: Olaya Rendueles

Reviewer(s): The reviewers have opted to remain anonymous.

Transaction Report:

DOI: <https://doi.org/10.1128/spectrum.00738-24>

Re: Spectrum00738-24 (Guanabenz acetate, an antihypertensive drug repurposed as an inhibitor of Escherichia coli biofilm)

Dear Dr. Arakkaveetil Kabeer Farha:

Thank you for the privilege of reviewing your work. Below you will find my comments, instructions from the Spectrum editorial office, and the reviewer comments.

Revision Guidelines

Sincerely,
Olaya Rendueles
Editor
Microbiology Spectrum

Reviewer #1 (Comments for the Author):

In this manuscript, Farha and colleagues investigated the anti-biofilm effects of guanabenz acetate (GABE), an antihypertensive drug identified in a drug screen, in an E. coli strain using a series of functional studies. Their results showed that GABE decreased curli and cellulose production, which was associated with decreased biofilm formation. RNAseq analyses further indicated that GABE may inhibit biofilm formation via the transcriptional repression of biofilm-associated genes. The data

presented is generally convincing in terms of demonstrating the potential anti-biofilm effects of GABE. However, critical methodological details are missing to evaluate whether the experimental design is sound and whether the results presented are reproducible.

MAJOR POINTS:

Missing key methodological details:

1. Please describe what type of *E. coli* strain is AR3110 (i.e., clinical? From sepsis, gut, etc?).
2. What was GABE dissolved in? Was a vehicle control included for all experiments? If so, what was the vehicle control?
3. What were the growth conditions utilized for the growth curves, biofilm studies, and RNAseq analysis (i.e., medium, temperature, and oxygenation)? For the biofilm studies, at what time points were the various measurements / images taken? What time point was used for the RNAseq analyses?
4. Scale bars are missing in most images (B, D, E, H, I).
5. Information regarding the number of independent experiments and sample size are missing for most experiments.
6. In the crystal violet assay data presented in F, it's unclear if the data in G is from the same assay, representing the total OD values from the same biofilm experiment.

Statistical analyses:

1. In F, it is not stated what statistical test was utilized or what the pairwise comparisons are (i.e., is 500 significant relative to 0?).
2. Stats are missing in C and G.
3. In M, it is not stated what statistical test was utilized or what the pairwise comparisons are.

Missing data / controls:

1. The supplemental data does not include the entire list of significant DEGs from the RNAseq experiment.
2. The Western blots presented in S4 lack loading or housekeeping controls. In addition, it's unclear if the time points / growth conditions are the same as the qRT-PCR data presented in M.

Overinterpretation of results: In line 136, the authors state that "As a non-antibiotic drug, GABE would not contribute to long-term antibiotic resistance." I think it is important to define what an antibiotic drug means in this context (i.e., exhibiting no growth defects? If so, GABE does at higher concentrations and even some modest effects on growth at the anti-biofilm concentrations). Furthermore, long term resistance to anti-biofilm agents is also possible through similar mechanisms of antibiotic resistance such as evolution of the drug target, increased expression of efflux pumps, etc.

Missing citation: In lines 39-40 under Importance, the authors indicate that "studies have shown that Guanabenz acetate's mode of action is to inhibit the synthesis of cellulose and curli amyloid protein." However, there is no further information provided regarding these studies in the main text, nor any citations referring to these studies.

MODERATE POINTS:

1. The MIC data described in lines 83-84 should either be cited if published elsewhere, or this data should be included in this manuscript.
2. Based on the data presented in 1D, it looks like GABE may have a much stronger growth inhibitory effect during bacterial growth on surfaces versus during planktonic growth. This is evident by the radius of the colonies in 1D. This should be pointed out in the manuscript text. It may also be useful to show time course data for growth on the calcofluor media by measuring the radius of each inoculum. Finally, is this specific to the calcofluor media, or were similar results observed in the Congo red medium and in regular LB medium with GABE?

Reviewer #2 (Comments for the Author):

General comment:

In the present work, authors described the effect of guanabenz acetate (GABE) on E.coli biofilm in vitro. GABE was able to inhibit several biofilm-associated phenotypes, particularly, microcolony morphology, pellicle and submerged biofilm formation, and EPS-associated gene expression. The state-of-the-art techniques used in the study provide valuable readouts for the initial screening of the repurposed drug against bacterial biofilm. However, some results and interpretations need revision. Please find my comments below.

Major comments:

1. Unfortunately, there is no description of methods used in this study. Therefore, it is complicated to estimate what was the experimental set up, why slightly different concentrations of GABE have been used throughout the experiments, what were the controls, GABE solvent, and which parameters have been recorded for the final conclusions. What is the rationale behinds showing W3110 strain ones in Fig. 1B? Please revise.
2. Please confirm the negative effect of GABE on bacterial viability with proper viability tests, e.g. determination of CFU/ml or spot assay in liquid culture and in biofilm, or live/dead staining followed by confocal imaging of the treated biofilm.
3. Please elaborate more on 1) potential mechanism of GABE inhibitory effect (direct interaction with transcriptional regulators, multitargeted effect and, therefore, systemic inhibitory effect on bacterial physiology, etc); 2) what are the role of upregulated genes found in RNA transcriptomics analysis; 3) what are the potential side effects of an antihypertensive drug repurposed for bacterial infection treatment.
4. Missing statistical analysis details, e.g. sample size, replications, significance in 1G.
5. Missing controls: biofilm/growth assays: any known antibiofilm/antibacterial compound against tested strains; ThT assay: values for GABE+ThT, ThT alone.
6. Figure legends are very limited. Some panels need more detailed explanation, e.g. staining used, scale bars, axis labels.

Dear Editor,

Thank you for giving us the opportunity to submit a revised draft of the manuscript “Guanabenz acetate, an antihypertensive drug repurposed as an inhibitor of *Escherichia coli* biofilm” for publication in Microbiology Spectrum Journal (Spectrum00738-24). We appreciate the time and effort that you and the reviewers dedicated to providing feedback on our manuscript. The suggestions offered by the reviewers have been immensely helpful. Most of the reviewers' suggestions have been implemented. Those changes are highlighted within the manuscript. Please see below for a point-by-point response to the reviewers' comments and concerns. Every line number refer to the revised manuscript file (Spectrum00738-24-Manuscript_Clean Version.doc) with no tracked changes.

We hope the revised manuscript will better suit Microbiology Spectrum Journal, and we thank you for your continued interest in our research.

Sincerely,

Farha Arakkaveettil Kabeer

Reviewer #1 (Comments for the Author):

In this manuscript, Farha and colleagues investigated the anti-biofilm effects of guanabenz acetate (GABE), an antihypertensive drug identified in a drug screen, in an *E. coli* strain using a series of functional studies. Their results showed that GABE decreased curli and cellulose production, which was associated with decreased biofilm formation. RNAseq analyses further indicated that GABE may inhibit biofilm formation via the transcriptional repression of biofilm-associated genes. The data presented is generally convincing in terms of demonstrating the potential anti-biofilm effects of GABE. However, critical methodological details are missing to evaluate whether the experimental design is sound and whether the results presented are reproducible.

Response: Thank you! We found your comments extremely helpful and have revised accordingly.

MAJOR POINTS:

Missing key methodological details:

1. Please describe what type of *E. coli* strain is AR3110 (i.e., clinical? From sepsis, gut, etc?).

Response: *E. coli* AR3110 is a derivative of non-pathogenic *E. coli* K-12 strain W3110 with restored ability to synthesize pEtN-cellulose, forms large and flat curli- and pEtN-cellulose-containing macrocolonies with long radial ridges and small wrinkles that also stain dark red with Congo red (Line 68-72).

2. What was GABE dissolved in? Was a vehicle control included for all experiments? If so, what was the vehicle control?

Response: GABE was dissolved in 100 % DMSO. The stock solution was prepared by dissolving 50 mg GABE in 1 mL DMSO (171.74 mM). Vehicle control (DMSO) was not included in any experiments. Up to 1% DMSO does not affect the biofilm development in *E. coli* AR3110, as mentioned by Pruteanu et al. (2020). In our experiments, the final concentration of DMSO was only 0.1% (Line 176-177)

We also confirmed the effect of DMSO on *E. coli* AR3110 bacterial growth. Incubation with DMSO at a concentration of 1% for 24 h had no effect on bacterial growth.

3. What were the growth conditions utilized for the growth curves, biofilm studies, and RNAseq analysis (i.e., medium, temperature, and oxygenation)? For the biofilm studies, at what time points were the various measurements / images taken? What time point was used for the RNAseq analyses?

Response: As suggested by the reviewer, we have included growth conditions and GABE drug preparation in the caption to the Figure. As our article is a short communication, we cannot add detailed methodology in the main text (Lines 171-219).

4. Scale bars are missing in most images (B, D, E, H, I).

Response: Thank you for pointing this out. We have replaced the images with a scale bar.

Image B: We have replaced the image with stereomicroscopy image (10x magnification). In order to identify the structure of bacterial colony in detail, images of colonies taken by iPhone 14 Promax camera are included in Supplementary data (Fig. S1).

Image D was taken by Gel imaging system (FUSION-FX7.EDGE, Vilber).

Image E was taken by iPhone 12.

Image H: scale bar was added in the images. A new image for control (image H) was provided

Image I: scale bar was added in the images.

5. Information regarding the number of independent experiments and sample size are missing for most experiments.

Response: We have added information regarding the number of independent experiments, sample size, and statistical analysis in Figure legend 1 (Lines 171-219).

6. In the crystal violet assay data presented in F, it's unclear if the data in G is from the same assay, representing the total OD values from the same biofilm experiment.

Response: Thank you for pointing this out. We have added a detailed protocol in the caption of the figure (Line 193-203).

Statistical analyses:

1. In F, it is not stated what statistical test was utilized or what the pairwise comparisons are (i.e., is 500 significant relative to 0?).

Response: We agree with the reviewer's assessment. Crystal violet staining assay was performed in four biological replicates and technical duplicates in each experiment. Statistical significance was determined with Kruskal-Wallis test followed by Dunn's multiple comparison test (Line 193-203).

2. Stats are missing in C and G.

Response: We agree with the reviewer's assessment. We have provided new graphs with statistical information.

3. In M, it is not stated what statistical test was utilized or what the pairwise comparisons are.

Response: We agree with the reviewer's assessment.

For real time PCR analysis (Figure 1N), we have replaced the image. Student's *t*-test was used to analyse the significant differences between control and treated samples. All experiments were performed independently three times. Data are expressed as mean \pm standard deviation with asterisks indicating the level of statistical significance: * $P < 0.05$, ** $P < 0.01$, and ns=non-significant (Line 212-219).

Missing data/Controls:

1. The supplemental data does not include the entire list of significant DEGs from the RNAseq experiment.

Response: Thank you for this suggestion. We have included an Excel file with the entire list of significant DEGs identified in RNA transcriptomic analysis in supplementary information. (Supplementary data 2).

2. The Western blots presented in S4 lack loading or housekeeping controls. In addition, it's unclear if the time points / growth conditions are the same as the qRT-PCR data presented in M.

Response: Thank you for this suggestion. The SDS-PAGE gel is shown as a sample loading control. Details of the experiment set up are included in the Figure legend Supporting Information (Line 249-256).

3. Overinterpretation of results: In line 136, the authors state that "As a non-antibiotic drug, GABE would not contribute to long-term antibiotic resistance." I think it is important to define what an antibiotic drug means in this context (i.e., exhibiting no growth defects? If so, GABE does at higher concentrations and even some modest effects on growth at the anti-biofilm concentrations). Furthermore, long term resistance to anti-biofilm agents is also possible through similar mechanisms of antibiotic resistance such as evolution of the drug target, increased expression of efflux pumps, etc.

Response: Thank you for pointing this out. The text has been corrected as follows:

GABE can exhibit antibiofilm action without inhibiting bacterial growth. Unlike traditional antibiotics, the drug's resistance will develop later if it has no effect on bacterial growth. Non-antibiotics are categorized as drugs that have demonstrated antimicrobial properties but are not developed to act as antibiotics or chemotherapeutics. In this context, GABE, a non-antibiotic drug that inhibits biofilms, would not contribute to the development of long-term antibiotic resistance (Line 139-145).

4. Missing citation: In lines 39-40 under Importance, the authors indicate that "studies have shown that Guanabenz acetate's mode of action is to inhibit the synthesis of cellulose and curli amyloid protein." However, there is no further information provided regarding these studies in the main text, nor any citations referring to these studies.

Response: Thank you for pointing this out. We have made necessary corrections in the text. The revised text reads as follows on Line 37-40.

MODERATE POINTS:

1. The MIC data described in lines 83-84 should either be cited if published elsewhere, or this data should be included in this manuscript.

Response: As suggested by the reviewer, we have included MIC methodology in main text. The growth inhibitory effect of GABE was not reported elsewhere (Line 83-85).

2. Based on the data presented in 1D, it looks like GABE may have a much stronger growth inhibitory effect during bacterial growth on surfaces versus during planktonic growth. This is evident by the radius of the colonies in 1D. This should be pointed out in the manuscript text. It may also be useful to show time course data for growth on the calcofluor media by measuring the radius of each inoculum. Finally, is this specific to the calcofluor media, or were similar results observed in the Congo red medium and in regular LB medium with GABE?

Response: We thank the reviewer for the excellent suggestion. GABE has shown a growth inhibitory effect on *E. coli* strains above 100 μ M concentration. This is evident in Congo Red agar plate assay and it is not due to the effect of Calcofluor stain. However, *E. coli* AR3110

biofilms form large and flat microcolonies on the agar, making its size larger than normal bacterial colonies (Pruteanu et al., 2020). We have included Figure S1 with a new image to compare the colony size after GABE exposure.

Reviewer #2 (Comments for the Author):

General comment:

In the present work, authors described the effect of guanabenz acetate (GABE) on *E. coli* biofilm in vitro. GABE was able to inhibit several biofilm-associated phenotypes, particularly, microcolony morphology, pellicle and submerged biofilm formation, and EPS-associated gene expression. The state-of-the-art techniques used in the study provide valuable readouts for the initial screening of the repurposed drug against bacterial biofilm. However, some results and interpretations need revision. Please find my comments below.

Response: Thank you for your thorough review and salient observations. We believe that the reviewer's comments help improve the overall quality of our manuscript.

Major comments:

1. Unfortunately, there is no description of methods used in this study. Therefore, it is complicated to estimate what was the experimental set up, why slightly different concentrations of GABE have been used throughout the experiments, what were the controls, GABE solvent, and which parameters have been recorded for the final conclusions. What is the rationale behinds showing W3110 strain ones in Fig. 1B? Please revise.

Response: Thank you for pointing this out. We have submitted our manuscript as Observation paper. Due to the word limit constraint, we cannot add a Methodology section to the article. As suggested by the reviewer, we have included growth conditions and experimental set up in Figure legend 1 (Line 171-219).

What is the rationale behinds showing W3110 strain ones in Fig. 1B?

Response: *E. coli* K-12 strain W3110 produces only amyloid curli fibers, which generate microcolony biofilms with a concentric ring pattern and dark red staining with CR (Serra et al., 2013b). we used this strain to confirm whether GABE has the potential to inhibit curli production in *E. coli* biofilm (Line 68-72).

2. Please confirm the negative effect of GABE on bacterial viability with proper viability tests, e.g. determination of CFU/ml or spot assay in liquid culture and in biofilm, or live/dead staining followed by confocal imaging of the treated biofilm.

Response: We thank the reviewer for this observation; the problem has been fixed. The data was included in supporting information (Figure S2).

3. Please elaborate more on 1) potential mechanism of GABE inhibitory effect (direct interaction with transcriptional regulators, multitargeted effect and, therefore, systemic inhibitory effect on bacterial physiology, etc); 3) what are the potential side effects of an antihypertensive drug repurposed for bacterial infection treatment.

Response: Thank you for this suggestion. It would have been interesting to explore this aspect. However, in our study, this would not have been possible because our aim was to identify whether GABE has an inhibitory effect on biofilm components in *E. coli*; we did not check the interaction of GABE on any other molecules, such as transcriptional regulators.

2) what are the role of upregulated genes found in RNA transcriptomics analysis;

Response: The primary elevated genes are primarily linked to *torCAD* operon and YbhFSR putative drug resistance exporter. The current work confirms prior reports of the overexpression of ATP-binding cassette (ABC) transporters and *torCAD* operon in biofilms following drug treatment (Line 119-123)

3) what are the potential side effects of an antihypertensive drug repurposed for bacterial infection treatment.

Response: It has been proposed that certain hypertension medications, including propranolol, doxazoin, and methyl-L-DOPA, can be effectively used as antibacterial medications in mice. Studies on the adverse effects of antihypertensive medications used to treat bacterial infections are limited. Even if GABE is already safe for human use, it should be necessary to reevaluate its pharmacological properties to develop new therapeutic applications (Line 145-152).

4. Missing statistical analysis details, e.g. sample size, replications, significance in 1G.

Response: We have updated the statistical analysis data.

5. Missing controls: biofilm/growth assays: any known antibiofilm/antibacterial compound against tested strains;

Response: Epigallocatechin gallate (200 µg/mL) was used as a positive control for the Congo red agar plate assay as a curli amyloid inhibitor (Figure S1).

6. ThT assay: values for GABE+ThT, ThT alone.

Response: Thank you for this suggestion. We have provided a new graph with GABE+ThT, ThT single data (Figure S3).

6. Figure legends are very limited. Some panels need more detailed explanation, e.g. staining used, scale bars, axis labels.

Response: As suggested by the reviewer, we have updated figure legend information (Line 171-219).

Re: Spectrum00738-24R1 (Guanabenz acetate, an antihypertensive drug repurposed as an inhibitor of Escherichia coli biofilm)

Dear Dr. Arakkaveetil Kabeer Farha:

Thank you for the privilege of reviewing your work. Your manuscript is close to being accepted.

I would urge you to tone down the claims that anti-biofilm drugs do not lead to resistance. Please use the wording "should", or "is expected to" and reference appropriately.

"Unlike traditional antibiotics, the drug's resistance will develop later if it has no effect on bacterial growth. Non-antibiotics are categorized as drugs that have demonstrated antimicrobial properties but are not developed to act as antibiotics or chemotherapeutics. In this context, GABE, a non-antibiotic drug that inhibits biofilms, would not contribute to the development of long-term antibiotic resistance"

Once this has modified, I will be able to accept your manuscript.

Please return the manuscript within 10 days; if you cannot complete the modification within this time period, please contact me. If you do not wish to modify the manuscript and prefer to submit it to another journal, notify me immediately so that the manuscript may be formally withdrawn from consideration by Spectrum.

Revision Guidelines

Sincerely,
Olaya Rendueles
Editor
Microbiology Spectrum

Dear Editor,

Thank you for giving us the opportunity to submit a revised draft of the manuscript “Guanabenz acetate, an antihypertensive drug repurposed as an inhibitor of *Escherichia coli* biofilm” for publication in Microbiology Spectrum Journal (Spectrum00738-24). We appreciate the time and effort that you and the reviewers dedicated to providing feedback on our manuscript. The suggestions offered by the reviewers have been immensely helpful. Most of the reviewers' suggestions have been implemented. Those changes are highlighted within the manuscript. Please see below for a point-by-point response to the reviewers' comments and concerns. Every line number refer to the revised manuscript file (Spectrum00738-24-Manuscript_Clean Version.doc) with no tracked changes.

We hope the revised manuscript will better suit Microbiology Spectrum Journal, and we thank you for your continued interest in our research.

Sincerely,

Farha Arakkaveettil Kabeer

Reviewer #1 (Comments for the Author):

In this manuscript, Farha and colleagues investigated the anti-biofilm effects of guanabenz acetate (GABE), an antihypertensive drug identified in a drug screen, in an *E. coli* strain using a series of functional studies. Their results showed that GABE decreased curli and cellulose production, which was associated with decreased biofilm formation. RNAseq analyses further indicated that GABE may inhibit biofilm formation via the transcriptional repression of biofilm-associated genes. The data presented is generally convincing in terms of demonstrating the potential anti-biofilm effects of GABE. However, critical methodological details are missing to evaluate whether the experimental design is sound and whether the results presented are reproducible.

Response: Thank you! We found your comments extremely helpful and have revised accordingly.

MAJOR POINTS:

Missing key methodological details:

1. Please describe what type of *E. coli* strain is AR3110 (i.e., clinical? From sepsis, gut, etc?).

Response: *E. coli* AR3110 is a derivative of non-pathogenic *E. coli* K-12 strain W3110 with restored ability to synthesize pEtN-cellulose, forms large and flat curli- and pEtN-cellulose-containing macrocolonies with long radial ridges and small wrinkles that also stain dark red with Congo red (Line 68-72).

2. What was GABE dissolved in? Was a vehicle control included for all experiments? If so, what was the vehicle control?

Response: GABE was dissolved in 100 % DMSO. The stock solution was prepared by dissolving 50 mg GABE in 1 mL DMSO (171.74 mM). Vehicle control (DMSO) was not included in any experiments. Up to 1% DMSO does not affect the biofilm development in *E. coli* AR3110, as mentioned by Pruteanu et al. (2020). In our experiments, the final concentration of DMSO was only 0.1% (Line 176-177)

We also confirmed the effect of DMSO on *E. coli* AR3110 bacterial growth. Incubation with DMSO at a concentration of 1% for 24 h had no effect on bacterial growth.

3. What were the growth conditions utilized for the growth curves, biofilm studies, and RNAseq analysis (i.e., medium, temperature, and oxygenation)? For the biofilm studies, at what time points were the various measurements / images taken? What time point was used for the RNAseq analyses?

Response: As suggested by the reviewer, we have included growth conditions and GABE drug preparation in the caption to the Figure. As our article is a short communication, we cannot add detailed methodology in the main text (Lines 171-219).

4. Scale bars are missing in most images (B, D, E, H, I).

Response: Thank you for pointing this out. We have replaced the images with a scale bar.

Image B: We have replaced the image with stereomicroscopy image (10x magnification). In order to identify the structure of bacterial colony in detail, images of colonies taken by iPhone 14 Promax camera are included in Supplementary data (Fig. S1).

Image D was taken by Gel imaging system (FUSION-FX7.EDGE, Vilber).

Image E was taken by iPhone 12.

Image H: scale bar was added in the images. A new image for control (image H) was provided

Image I: scale bar was added in the images.

5. Information regarding the number of independent experiments and sample size are missing for most experiments.

Response: We have added information regarding the number of independent experiments, sample size, and statistical analysis in Figure legend 1 (Lines 171-219).

6. In the crystal violet assay data presented in F, it's unclear if the data in G is from the same assay, representing the total OD values from the same biofilm experiment.

Response: Thank you for pointing this out. We have added a detailed protocol in the caption of the figure (Line 193-203).

Statistical analyses:

1. In F, it is not stated what statistical test was utilized or what the pairwise comparisons are (i.e., is 500 significant relative to 0?).

Response: We agree with the reviewer's assessment. Crystal violet staining assay was performed in four biological replicates and technical duplicates in each experiment. Statistical significance was determined with Kruskal-Wallis test followed by Dunn's multiple comparison test (Line 193-203).

2. Stats are missing in C and G.

Response: We agree with the reviewer's assessment. We have provided new graphs with statistical information.

3. In M, it is not stated what statistical test was utilized or what the pairwise comparisons are.

Response: We agree with the reviewer's assessment.

For real time PCR analysis (Figure 1N), we have replaced the image. Student's *t*-test was used to analyse the significant differences between control and treated samples. All experiments were performed independently three times. Data are expressed as mean \pm standard deviation with asterisks indicating the level of statistical significance: * $P < 0.05$, ** $P < 0.01$, and ns=non-significant (Line 212-219).

Missing data/Controls:

1. The supplemental data does not include the entire list of significant DEGs from the RNAseq experiment.

Response: Thank you for this suggestion. We have included an Excel file with the entire list of significant DEGs identified in RNA transcriptomic analysis in supplementary information. (Supplementary data 2).

2. The Western blots presented in S4 lack loading or housekeeping controls. In addition, it's unclear if the time points / growth conditions are the same as the qRT-PCR data presented in M.

Response: Thank you for this suggestion. The SDS-PAGE gel is shown as a sample loading control. Details of the experiment set up are included in the Figure legend Supporting Information (Line 249-256).

3. Overinterpretation of results: In line 136, the authors state that "As a non-antibiotic drug, GABE would not contribute to long-term antibiotic resistance." I think it is important to define what an antibiotic drug means in this context (i.e., exhibiting no growth defects? If so, GABE does at higher concentrations and even some modest effects on growth at the anti-biofilm concentrations). Furthermore, long term resistance to anti-biofilm agents is also possible through similar mechanisms of antibiotic resistance such as evolution of the drug target, increased expression of efflux pumps, etc.

Response: Thank you for pointing this out. The text has been corrected as follows:

GABE can exhibit antibiofilm action without inhibiting bacterial growth. Unlike traditional antibiotics, the drug's resistance will develop later if it has no effect on bacterial growth. Non-antibiotics are categorized as drugs that have demonstrated antimicrobial properties but are not developed to act as antibiotics or chemotherapeutics. In this context, GABE, a non-antibiotic drug that inhibits biofilms, would not contribute to the development of long-term antibiotic resistance (Line 139-145).

4. Missing citation: In lines 39-40 under Importance, the authors indicate that "studies have shown that Guanabenz acetate's mode of action is to inhibit the synthesis of cellulose and curli amyloid protein." However, there is no further information provided regarding these studies in the main text, nor any citations referring to these studies.

Response: Thank you for pointing this out. We have made necessary corrections in the text. The revised text reads as follows on Line 37-40.

MODERATE POINTS:

1. The MIC data described in lines 83-84 should either be cited if published elsewhere, or this data should be included in this manuscript.

Response: As suggested by the reviewer, we have included MIC methodology in main text. The growth inhibitory effect of GABE was not reported elsewhere (Line 83-85).

2. Based on the data presented in 1D, it looks like GABE may have a much stronger growth inhibitory effect during bacterial growth on surfaces versus during planktonic growth. This is evident by the radius of the colonies in 1D. This should be pointed out in the manuscript text. It may also be useful to show time course data for growth on the calcofluor media by measuring the radius of each inoculum. Finally, is this specific to the calcofluor media, or were similar results observed in the Congo red medium and in regular LB medium with GABE?

Response: We thank the reviewer for the excellent suggestion. GABE has shown a growth inhibitory effect on *E. coli* strains above 100 μ M concentration. This is evident in Congo Red agar plate assay and it is not due to the effect of Calcofluor stain. However, *E. coli* AR3110

biofilms form large and flat microcolonies on the agar, making its size larger than normal bacterial colonies (Pruteanu et al., 2020). We have included Figure S1 with a new image to compare the colony size after GABE exposure.

Reviewer #2 (Comments for the Author):

General comment:

In the present work, authors described the effect of guanabenz acetate (GABE) on *E. coli* biofilm in vitro. GABE was able to inhibit several biofilm-associated phenotypes, particularly, microcolony morphology, pellicle and submerged biofilm formation, and EPS-associated gene expression. The state-of-the-art techniques used in the study provide valuable readouts for the initial screening of the repurposed drug against bacterial biofilm. However, some results and interpretations need revision. Please find my comments below.

Response: Thank you for your thorough review and salient observations. We believe that the reviewer's comments help improve the overall quality of our manuscript.

Major comments:

1. Unfortunately, there is no description of methods used in this study. Therefore, it is complicated to estimate what was the experimental set up, why slightly different concentrations of GABE have been used throughout the experiments, what were the controls, GABE solvent, and which parameters have been recorded for the final conclusions. What is the rationale behinds showing W3110 strain ones in Fig. 1B? Please revise.

Response: Thank you for pointing this out. We have submitted our manuscript as Observation paper. Due to the word limit constraint, we cannot add a Methodology section to the article. As suggested by the reviewer, we have included growth conditions and experimental set up in Figure legend 1 (Line 171-219).

What is the rationale behinds showing W3110 strain ones in Fig. 1B?

Response: *E. coli* K-12 strain W3110 produces only amyloid curli fibers, which generate microcolony biofilms with a concentric ring pattern and dark red staining with CR (Serra et al., 2013b). we used this strain to confirm whether GABE has the potential to inhibit curli production in *E. coli* biofilm (Line 68-72).

2. Please confirm the negative effect of GABE on bacterial viability with proper viability tests, e.g. determination of CFU/ml or spot assay in liquid culture and in biofilm, or live/dead staining followed by confocal imaging of the treated biofilm.

Response: We thank the reviewer for this observation; the problem has been fixed. The data was included in supporting information (Figure S2).

3. Please elaborate more on 1) potential mechanism of GABE inhibitory effect (direct interaction with transcriptional regulators, multitargeted effect and, therefore, systemic inhibitory effect on bacterial physiology, etc); 3) what are the potential side effects of an antihypertensive drug repurposed for bacterial infection treatment.

Response: Thank you for this suggestion. It would have been interesting to explore this aspect. However, in our study, this would not have been possible because our aim was to identify whether GABE has an inhibitory effect on biofilm components in *E. coli*; we did not check the interaction of GABE on any other molecules, such as transcriptional regulators.

2) what are the role of upregulated genes found in RNA transcriptomics analysis;

Response: The primary elevated genes are primarily linked to *torCAD* operon and YbhFSR putative drug resistance exporter. The current work confirms prior reports of the overexpression of ATP-binding cassette (ABC) transporters and *torCAD* operon in biofilms following drug treatment (Line 119-123)

3) what are the potential side effects of an antihypertensive drug repurposed for bacterial infection treatment.

Response: It has been proposed that certain hypertension medications, including propranolol, doxazoin, and methyl-L-DOPA, can be effectively used as antibacterial medications in mice. Studies on the adverse effects of antihypertensive medications used to treat bacterial infections are limited. Even if GABE is already safe for human use, it should be necessary to reevaluate its pharmacological properties to develop new therapeutic applications (Line 145-152).

4. Missing statistical analysis details, e.g. sample size, replications, significance in 1G.

Response: We have updated the statistical analysis data.

5. Missing controls: biofilm/growth assays: any known antibiofilm/antibacterial compound against tested strains;

Response: Epigallocatechin gallate (200 µg/mL) was used as a positive control for the Congo red agar plate assay as a curli amyloid inhibitor (Figure S1).

6. ThT assay: values for GABE+ThT, ThT alone.

Response: Thank you for this suggestion. We have provided a new graph with GABE+ThT, ThT single data (Figure S3).

6. Figure legends are very limited. Some panels need more detailed explanation, e.g. staining used, scale bars, axis labels.

Response: As suggested by the reviewer, we have updated figure legend information (Line 171-219).

Dear Editor,

Thank you for giving us the opportunity to submit a revised draft of the manuscript “Guanabenz acetate, an antihypertensive drug repurposed as an inhibitor of *Escherichia coli* biofilm” for publication in Microbiology Spectrum Journal (Spectrum00738-24). **As suggested by the editor, we have made necessary corrections in the text.** Those changes are highlighted within the manuscript. Every line number refer to the revised manuscript file (Spectrum00738-24-Manuscript_Clean Version.doc) with no tracked changes.

We hope the revised manuscript will better suit Microbiology Spectrum Journal, and we thank you for your continued interest in our research.

Sincerely,

Farha Arakkaveettil Kabeer

Comment:

I would urge you to tone down the claims that anti-biofilm drugs do not lead to resistance. Please use the wording "should", or "is expected to" and reference appropriately.

"Unlike traditional antibiotics, the drug's resistance will develop later if it has no effect on bacterial growth. Non-antibiotics are categorized as drugs that have demonstrated antimicrobial properties but are not developed to act as antibiotics or chemotherapeutics. In this context, GABE, a non-antibiotic drug that inhibits biofilms, would not contribute to the development of long-term antibiotic resistance"

Response: Thank you! We found your comment extremely helpful and have revised accordingly. The text has been corrected as follows:

Drugs with antimicrobial activity but not intended for use as antibiotics are classified as non-antibiotics (22). The absence of the bactericidal effect of an antibiofilm agent reduces the selective pressure against biofilm development, hence reducing the likelihood of bacteria developing resistance to it (23). At lower doses, GABE can demonstrate antibiofilm effect without impeding bacterial growth. In this context, it is anticipated that bacteria will be less likely to become resistant to GABE, a non-antibiotic drug that inhibits biofilms. (Line 140-146).

Re: Spectrum00738-24R2 (Guanabenz acetate, an antihypertensive drug repurposed as an inhibitor of Escherichia coli biofilm)

Dear Dr. Arakkaveetil Kabeer Farha:

Your manuscript has been accepted, and I am forwarding it to the ASM production staff for publication. Your paper will first be checked to make sure all elements meet the technical requirements. ASM staff will contact you if anything needs to be revised before copyediting and production can begin. Otherwise, you will be notified when your proofs are ready to be viewed.

Sincerely,
Olaya Rendueles
Editor
Microbiology Spectrum